# *Flaviviridae* Nonstructural Proteins: The Role in Molecular Mechanisms of Triggering Inflammation

**DOI:** 10.3390/v14081808

**Published:** 2022-08-18

**Authors:** Anastasia Latanova, Elizaveta Starodubova, Vadim Karpov

**Affiliations:** Engelhardt Institute of Molecular Biology, Russian Academy of Sciences, 119991 Moscow, Russia

**Keywords:** *Flaviviridae*, nonstructural proteins, innate immunity, inflammasome, inflammation, TLR signaling, RLR signaling, NLR signaling, NF-κB, STING, cGAS

## Abstract

Members of the *Flaviviridae* family are posing a significant threat to human health worldwide. Many flaviviruses are capable of inducing severe inflammation in humans. *Flaviviridae* nonstructural proteins, apart from their canonical roles in viral replication, have noncanonical functions strongly affecting antiviral innate immunity. Among these functions, antagonism of type I IFN is the most investigated; meanwhile, more data are accumulated on their role in the other pathways of innate response. This review systematizes the last known data on the role of *Flaviviridae* nonstructural proteins in molecular mechanisms of triggering inflammation, with an emphasis on their interactions with TLRs and RLRs, interference with NF-κB and cGAS-STING signaling, and activation of inflammasomes.

## 1. Introduction

*Flaviviridae* is a family of enveloped ss(+RNA) viruses, consisting of four genera: *Flavivirus*, *Hepacivirus*, *Pegivirus*, and *Pestivirus*. Members of *Flavivirus* are mosquito-borne and tick-borne viruses, most of which infect humans, causing a spectrum of clinical symptoms from mild febrile illness to severe hemorrhagic fever or neuroinvasion. It includes Zika virus (ZIKV), West Nile virus (WNV), dengue virus (DENV), Yellow fever virus (YFV), Japanese encephalitis virus (JEV), tick-borne encephalitis virus (TBEV), and also non-mammal viruses such as Duck Tembusu virus (DTMUV). *Hepacivirus* includes bloodborne human Hepatitis C virus (HCV). Genus *Pestivirus* includes viruses that infect nonhuman mammals, such as bovine viral diarrhea virus (BVDV) and classical swine fever virus (CSFV). *Pegivirus* includes several viruses infecting different groups of mammals comprising humans.

Nowadays, *Flaviviridae* are widespread all over the world. DENV infection is endemic in more than 100 countries, and accounts for approximately 400 million infections per year with 96 million exhibiting a severe disease condition, and thus poses a major public problem [1]. The number of persons chronically infected with HCV is estimated at 58 million [2]. These viruses, and also WNV, JEV ZIKV, and YFV, account for the most cases of infections by *Flaviviridae*. The burden of flaviviruses is also complicated by the fact that the prophylactic vaccines are licensed only for DENV [3], JEV [4], YFV [5], and TBEV [6], and are not universally effective and safe. The effectiveness of a licensed live-attenuated vaccine against DENV, Dengvaxia^®^, differs between DENV serotypes, either due to residual differences in antigen persistence between chimeric vaccine strains or due to prior anti-DENV immunity, which induces an antibody-dependent enhancement of infection [7]. Furthermore, except for HCV antivirals [7], no specific therapy against the proteins of other *Flaviviridae* family members is available for clinical use.

The symptoms of human *Flaviviridae* infections very often include significant systemic inflammatory reactions. Generally, the common mechanism of inflammation implies four steps: sensing of damage-/pathogen-associated molecular patterns (DAMPs/PAMPs) by PRRs (pattern recognition receptors); triggering of molecular signaling cascades, such as NF-κB, cGAS-STING, and IFN pathways; the release of proinflammatory cytokines and chemokines; and the recruitment of inflammatory cells [8]. Severe forms of DENV infection are characterized by the excessive release of proinflammatory cytokines and chemokines, such as TNF-α, IFN-γ, IL-6, IL-1β, MIF (macrophage migration inhibitory factor), IL-8, CXCL10, and CCL2 [9,10,11,12] which leads to the development of a cytokine storm accompanied by dengue hemorrhagic fever and dengue shock syndrome [13,14], and is hypothesized to be a mechanism for vascular leak [15,16,17,18,19]. ZIKV infection is also highly inflammatory, inducing a cytokine profile similar to a profile induced by DENV infection, and upregulation of cellular adhesion molecules [20,21,22]. In addition, neurotropic viruses such as ZIKV, JEV, TBEV, and non-neurotropic DENV and YFV cause neuroinflammation, by releasing proinflammatory factors and causing the infiltration of immune cells into the brain [20,23,24,25,26,27]. Finally, hepatic inflammation is caused by YFV and HCV [28,29], and in the case of HCV it may lead to hepatic fibrosis, liver injury, and cirrhosis [29]. Thus, inflammation is a burden of many *Flaviviridae* infections.

It is well studied that inflammatory reactions are triggered by the propagation of *Flaviviridae*. *Flaviviridae* RNA genome is approximately 10–11 kb and is represented by a long ORF flanked by 3′ and 5′-UTR with a 5′-end cap [30,31]. The single ORF is translated into a single polyprotein, which is processed by viral protease NS2B/NS3 and cellular proteases [32,33] into three structural proteins comprising the viral particle, C; prM and E protein; and seven nonstructural proteins, playing the role in viral replication, namely, NS1, NS2A, NS2B, NS3, NS4A, NS4B, and NS5 [34].

Structural flaviviral proteins build up viral particles and play the role in antiviral immune response. E protein is a glycoprotein of 53 kDa, forming the shell of flaviviruses, responsible for binding of the virus to the receptors [35,36], and is also the principal target of immune response, which elicits neutralizing antibodies [37]. C protein has a molecular weight of approximately 12 kDa, has the affinity for viral RNA and lipid membranes, and forms the part of the nucleocapsid [38,39]. PrM protein weighs approximately 19–21 kDa and is required for the prevention of premature fusion of the E protein within acidic compartments, serving as a chaperone for E protein, and facilitating translocation into the ER lumen during polyprotein translation [40,41,42].

Nonstructural proteins of flaviviruses, though not being a part of viral particles, play different roles in the viral life cycle, and have multiple functions beyond this. NS1 is a 40–50 kDa glycoprotein, depending on the glycosylation status, which is cleaved from NS2A [43], and translocated to the ER lumen, where it is co-translationally processed by a host signal peptidase [44]. It may be expressed within intracellular membranes, or at the cell surface [45,46]. Additionally, it is colocalized with dsRNA and interacts with NS4A and NS4B, and thus may be a component of replicase [47,48,49]. NS2A is a 22–25 kDa transmembrane protein [50,51]. Its N-terminus is processed in the ER lumen by a host protease and C-terminus in the cytoplasm by viral protease NS2B/NS3 [44,52]. NS2A is shown to play different roles depending on the *Flaviviridae* species, and may be essential in the viral life cycle processes, including RNA synthesis, the formation of the replication complex, and virion assembly/maturation/secretion [53]. It also may act as a viroporin and be involved in transporting viral RNA from sites of RNA replication, across virus-induced membranes, to adjacent sites of virus assembly [54,55]. NS2B is an integral membrane protein of 14 kDa, with a central hydrophilic region necessary for the activation of NS3 protease, and hydrophobic domains which may be involved in the oligomerization of the viral protease [56]. NS2B interacts with the NS2A protein, thus playing an important role in viral replication and assembly [57]. It also may act as a viroporin similar to NS2A [58,59]. NS2A and NS2B of DENV and JEV have been shown to exhibit viroporin activities such as the abilities to oligomerize, produce pore-like structures, and modify the permeability of the membranes, which lead to membrane destabilization and may be critical in the viral replication cycle [58,59,60]. NS3 is a 69 kDa protein, with a N-terminal protease domain with chymotrypsin-like activity and is able to perform *cis* and *trans* cleavage of the viral polyprotein, generating nonstructural proteins, and the C-terminal NTPase/RNA helicase. The NS2B cofactor is necessary for NS3 protease activity [61]. NS4A has a molecular mass of approximately 16 kDa [62], has a transmembrane domain, and plays a role in the membrane rearrangement supporting viral replication complexes [63]. It also may interact with other nonstructural proteins including NS1, NS2A, and NS5 [64]. It plays a crucial role during flaviviral replication, by inducing autophagy, as shown in DENV and ZIKV viruses [65,66]. NS4B is a 27 kDa protein in its mature form, with transmembrane domains [67,68], which participates in the assembly of the viral replication complex, in particular through interaction with NS1, NS3, and NS4A [49,69]. NS5 is the largest and the most conserved flaviviral protein, with a molecular weight of approximately 105 kDa [70]. Its N-terminal domain has a methyltransferase (MTase) activity responsible for the viral cap formation [71], and its C-terminus is presented by an RNA-dependent RNA polymerase (RdRp) domain responsible for the genome replication [72]. Viral replication also requires the formation of an NS5-NS3 complex [73,74]. In addition, NS5 is expected to have additional functions not yet identified.

The key factors involved in triggering inflammatory and antiviral responses are PRRs. They include Toll-like receptors (TLRs), RIG-I-like receptors (RLRs), NOD-like receptors (NLRs), AIM2-like receptors, C-type lectin receptors, and cytoplasmic DNA and RNA sensors [75,76,77]. The recognition of DAMPs and PAMPs by PRRs induces a variety of signaling cascades, including NF-κB, type I IFN response, and inflammasome activation, which lead to the production of proinflammatory and antiviral cytokines and chemokines [78]. Virus-associated DNA and RNA, and dsRNA produced by infected cells are classical viral PAMPs sensed by PRRs [78]. However, over the last years, more data have accumulated noting that proteins of different viruses including *Flaviviridae* interact with PRRs acting as PAMPs and may interfere with downstream signaling pathways. Nonstructural proteins of flaviviruses, besides their roles in the viral life cycle, interact with host cells and interfere with cellular functions. They can activate or inhibit immune signaling pathways leading to the activation or abrogation of inflammatory and antiviral responses. In this review, *Flaviviridae* nonstructural protein interactions with TLRs, RLRs, cytoplasmic DNA sensors, and NLRs, as well as their interference with downstream cascades, are discussed. A discussion of the JAK-STAT signaling pathway in view of *Flaviviridae* nonstructural proteins is omitted, as this aspect is covered by several recent reviews [79,80,81,82]. The mechanisms of interference are presented in several tables classified by signaling cascades, accompanied with a short theoretical introduction for every cascade, comments on flaviviral protein interactions, and a discussion.

## 2. TLRs and *Flaviviridae* Nonstructural Proteins

### 2.1. TLR Signaling

TLR is a type I transmembrane glycoprotein consisting of three domains: N-terminal extracellular domain with a variable number of leucine-rich repeats (LRRs) motifs, which is responsible for the recognition of a pathogen; a transmembrane domain; and a cytoplasmic Toll-interleukin 1 receptor (TIR) homology domain, for the recruitment of adaptor proteins, such as MyD88 (myeloid differentiation primary response 88) and TRIF (TIR domain-containing adapter inducing IFN-β) [83,84]. Currently, TLR1-TLR10 are known in humans, and of them, TLR1-TLR9 are conserved in both humans and mice, though analogous human and murine TLRs have a number of structural differences [85]. TLRs differ by their cellular localization and ligand recognition and are mainly expressed by antigen presenting cells (macrophages and dendritic cells), but also found in non-immune cells, such as fibroblasts and epithelial cells [86]. TLR1, TLR2, TLR4, TLR5, and TLR6 are cell surface receptors and mostly recognize the membrane components of different microorganisms such as lipids, proteins, and lipoproteins [87]. TLR2 and TLR4 are also involved in the recognition of viral envelope proteins on the cell surface and play a critical role in the recognition of bacterial components [88]. TLR3, TLR7, TLR8, and TLR9 have intracellular localization, locating in ER, endosomes, lysosomes, and endolysosomes [86]. There is also evidence that TLR3, TLR7, and TLR9 may be localized in the cell membrane [89]. TLR3 recognizes viral dsRNA, siRNAs, and self-RNAs, which are derived from damaged cells. A synthetic analog of viral dsDNA, used in many studies to mimic viral infection, which activates TLR3, is poly(I:C) (polyinosinic-cytidylic acid) [87]. Not only can RNA viruses be sensed by TLR3, but DNA viruses, such as HSV1, HSV2, and EBV are capable of generating RNA intermediates during viral replication, sensed by TLR3 as well [90,91,92]. TLR7 and TLR8 are involved in sensing ssRNA viruses [88,93]. TLR9 is responsible for the recognition of bacterial and viral DNA that is rich in CpG-DNA motifs [94,95]. Murine TLR10 is a pseudogene because of retrovirus insertion [87]; meanwhile, human TLR10 is functional in sensing influenza A infection and, in collaboration with TLR2, sensing bacterial ligands of listeria [86]. Furthermore, mice have TLR11, TLR12, and TLR13, absent in humans, which are involved in the recognition of different ligands of viral, bacterial, and protozoan origin [86].

The sensing of the ligands by TLRs stimulates two signaling cascades: the MyD88-dependent pathway, which mainly leads to the production of proinflammatory cytokines and chemokines; and TRIF-dependent pathway, which leads to the production of IFNs and proinflammatory cytokines [96]. The TRIF-dependent pathway is utilized by TLR3 and TLR4, and the MyD88-dependent by all other TLRs. TLR4 is the only receptor which triggers both MyD88- and TRIF-dependent pathways [97,98,99]. In the MyD88-dependent pathway, TLRs recruit MyD88 and MAL (also called TIRAP) proteins to their TIR domain, and in the TRIF-dependent pathway, TRIF (also called TICAM1) and TRAM (TRIF-related adaptor molecule, also called TICAM2) proteins are recruited [88].

In the MyD88-dependent pathway, after PAMPs sensing, TLRs homo- or heterodimerize and recruit MyD88, which forms a complex with IRAK1 and IRAK4 (IL-1 receptor associated kinase family members) [100]. IRAK4 activates IRAK1, which is sequentially auto-phosphorylated [101] and released from MyD88 [102]. The next step is the association of IRAK1 with RING-type E3 ubiquitin ligase TRAF6 (TNF receptor associated factor 6). TRAF6 mediates the activation of the TAK1 protein kinase complex (transforming growth factor-β-activated kinase 1) through promoting K63-polyubiquitination of itself and of TAK1 [86]. The TAK1 complex in its turn phosphorylates the IKK complex, which consists of two IKKs (IkB kinases), IKKα and IKKβ, and a regulatory subunit NEMO (NF-κB essential modulator, also called IKKγ) [103]. NEMO has the ability to bind to K63-linked polyubiquitin chains [104], thus playing an important role in NF-κB canonical activation. The IKK complex triggers the canonical mechanism of NF-κB (nuclear factor kappa-light-chain-enhancer of activated B cells) activation. NF-κB plays an essential role in the regulation of proinflammatory cytokines, interferons, and chemokines in virus-infected cells [105,106]. Inactive NF-κB is located in the cytoplasm where it is associated with IκBα, which masks NF-κB nuclear localization signals (NLS) [107]. The IKK complex triggered by TLR activation targets IκBα for proteasome degradation by phosphorylation [108]. This results in the translocation of NF-κB to the nucleus [109,110], where it binds to DNA-binding sites of specific genes and thus triggers their expression and the production of proinflammatory cytokines [78]. NF-κB may be activated not only through the TLR pathway, but also in response to various cytokines, i.e., to TNF-α or IL-1β binding to their receptors [111]. The activation of NF-κB by TNF-α requires the interaction of TNF-α with TNFR1 (tumor necrosis factor receptor 1), and the recruitment of TNFR1 with TRADD, RIP1, and TRAF2 proteins to plasma membrane lipid rafts [112].

In specific immune cells subsets, such as plasmacytoid dendritic cells, TLR7- and TLR9-MyD88-dependent signaling can trigger the production of type I IFNs. This requires the recruitment of TRAF3 (TNF receptor associated factor 3) and IKKα to the MyD88-IRAK-TRAF6 complex. Then, IRAK1 and IKKα phosphorylate transcription factor IRF7 (interferon regulatory factor 7) [96], which translocates to the nucleus and binds to ISRE (IFN-stimulated response elements), thus inducing the expression of type I IFN [113]. Secreted type I IFNs bind to IFNAR1/IFNAR2 (type I/type II interferon receptor) and induce the JAK-STAT signaling pathway and expression of IFN-stimulated genes (ISGs), thus inducing an antiviral state [114]. TAK1 can also activate another inflammatory signaling cascade: a mitogen-activated protein kinase (MAPK). TAK1 phosphorylates MKK3/MKK6, which subsequently phosphorylates several protein kinases, including ERK1/2 (extracellular signal regulated kinases 1/2), JNK (c-Jun N terminal kinases), and p38 (p38 mitogen-activated protein kinase). These kinases mediate the activation of AP-1 (activator protein 1) transcription factor family proteins, which regulate the production of proinflammatory cytokines [86,87,114].

The activation of TLR3/TLR4 results in the interaction of TRIF with a multiprotein signaling complex consisting of TRAF6, TRADD, Pellino-1 (a member of the Pellino family E3 ubiquitin ligases), and RIP1 (receptor-interacting protein 1), which activates the TAK1 and IKK complex, and, finally MAPK and NF-κB [87,115]. In addition to NF-κB activation, the TRIF-dependent pathway also activates IRF3, and type I IFN, as TRIF, with a participation of TRAF3 recruits a signaling complex of non-canonical IKK-related kinases, TBK1 (TANK binding kinase 1), and IKKε (also called IKKi), which phosphorylate IRF3 and also IRF7, and thus activate IFN production [87].

Thus, both MyD88-dependent and TRIF-dependent TLR signaling result in the activation of NF-κB and IFN responses, by different mechanisms.

### 2.2. Flaviviridae Nonstructural Proteins’ Interference with TLR Signaling

TLRs sense the dsRNA of many viruses including *Flaviviridae*. TLR3 is a classic receptor for viral dsRNA and can recognize the dsRNA of ZIKV, WNV, DENV, ATMUV (Avian Tembusu Virus), and JEV [116,117,118,119,120,121,122]. TLR3 also senses HCV infection, and namely, the dsRNA intermediates produced during infection, and not structural HCV RNA [123]. Interestingly, HCV infection induces the abundance of TLR3 adaptor TRIF and impairs poly (I:C)-induced signaling [123]. TLR7 is known as a regulator of the expression of type I IFN and proinflammatory cytokines in response to ZIKV, WNV, DENV, JEV, HCV (and TLR8 also), and Langat virus infection [124,125,126,127,128,129,130]. At the same time, numerous interactions of flavivirus nonstructural proteins with TLRs and with downstream proteins of TLR signaling are known (Table 1).

DENV interference with TLR signaling is extensively studied. DENV, as described above, causes severe hemorrhagic fever, with vascular leakage, and the establishment of the reasons for this is the focus of many studies. DENV NS1 protein behaves as a virulence factor in many of these studies. NS1 exists in different cellular locations in several oligomeric forms, having different functions. During DENV infection, monomers of NS1 arrange in dimers in the ER, and these dimers remain membrane-associated, either in vesicular compartments within the cell or on the cell surface membrane [45]. Three dimeric forms of NS1 may arrange forming barrel-shaped hexamers [46], which can detach from the membrane and be secreted into the extracellular milieu in a form of soluble nanoparticles containing lipids [45]. Intracellular NS1 is essential in viral replication [131], and cell-surface associated and secreted extracellular NS1 plays an important role in immune response and viral pathogenesis. Secreted NS1 may be phagocytosed by different immune cells, including macrophages, dendritic cells, and monocytes and further activates them, and also binds to complement and coagulation factors, alters the glycocalyx barrier, promotes endothelial permeability, and induces inflammatory and adaptive immune responses [45,132,133,134,135,136,137,138]. DENV NS1 is shown to behave like LPS, being able to activate TLR4 and its downstream signaling and, subsequently, vascular leak, at least, in vitro [136] (Table 1). Interestingly, the other study demonstrated that there are some other TLR4-independent mechanisms of vascular leak activation, as even in TLR4-deficient mice, substantial vascular leak in endothelial cells, dependent on NS1, was still observed [139] (Table 1). In addition, the direct interaction of platelets TLR4 and NS1 was demonstrated which led to the release of platelets’ proinflammatory mediators and to the platelets’ adhesion to endothelial cells [140,141] (Table 1). DENV infection of platelets was shown to be abortive, but they could secrete NS1, which activated platelets in an autocrine manner [141]. Furthermore, a correlation was found between positivity for NS1 antigen and elevated levels of endocan, which is a highly specific biomarker of endothelial cell activation, in the serum of DENV-infected patients [142]. In vitro studies on a model of endothelial cells indicated that the NS1 protein of all four DENV serotypes could induce endocan expression by a TLR4-dependent mechanism [142]. The NS1 of other flaviviruses including ZIKV, WNV, JEV, and YFV was also demonstrated to bind to human endothelial cells by disrupting endothelial glycocalyx components, causing tissue-specific vascular leakage in mice [138].

However, in the case of DENV, there are also controversial data. For example, DENV-infected patients suffering from a more severe form of dengue with hemorrhagic fever in comparison to the patients with milder dengue, displayed higher NS1 serum levels correlated with low TLR4 expression and impaired TLR4 signaling during the acute febrile phase of the disease [143] (Table 1). This contradicts with the above-mentioned studies, where NS1 upregulated TLR4 signaling; thus, more studies to settle this argument are needed. In addition, one study indicated that TLR2 and TLR6 are activated by DENV NS1 [144]; meanwhile, most of the other studies regarded only TLR4 and not TLR2/6 NS1-dependent activation (Table 1). Modhiran et al. in their paper provided more experiments that TLR2/6 are not activated by NS1 [135], and pointed out that in the study by Chen J. et al. commercial *E. coli* derived NS1 was used, which exhibited wrong protein folding and was monomeric under all the conditions [135]. Additionally, contamination of the commercial NS1 by several microbial TLR ligands was detected, that could also non-specifically activate TLR2 and TLR6 [135]. Modhiran et al. concluded that the use of commercial bacterial-expressed NS1 for studies of innate immune responses is inadequate [135].

Anti-DENV antibodies to NS1 are also capable of exhibiting an interesting double-faced effect. From one side, a bunch of studies indicated that both vaccine/infection-derived and monoclonal anti-NS1 antibodies have protective activity in vivo [145,146,147,148,149,150,151,152,153,154,155]. From the other side, some studies demonstrated that anti-DENV NS1 antibodies cross-react with clotting factors, platelets, and endothelial cells, leading to the production of proinflammatory cytokines and endothelial cell damage [156,157,158,159,160]; however, the contributions of these cross-reactive antibodies in DENV pathogenesis in vivo are still under debate. Taken together, the development of monoclonal Abs and vaccines against DENV based on NS1 should be carefully performed, and the epitopes of NS1 able to elicit auto-antibodies in the organisms should be omitted [161,162].

Data on the WNV evasion of TLR signaling are concerned with the interaction of the WNV NS1 protein with TLR3 (Table 1). Several studies indicated that WNV NS1 can inhibit TLR3 signaling and also TLR4/7 signaling in macrophages and dendritic cells, and the production of proinflammatory cytokines [163,164,165] (Table 1). Secreted NS1 can bind to the surface of different cell types. In particular, it can bind to murine macrophages and dendritic cells, predominantly in the draining lymph nodes of mice, and inhibit IL-6 and IFN-β responses in vivo [164]. Direct binding of NS1 to TLR3 has not been shown [166], but it is speculated that NS1 could interfere with downstream signaling such as the recruitment of TRIF and activation of TBK1 [163]. Finally, no exact mechanism of how WNV NS1 inhibits TLR3 is known. Interestingly, while some studies indicated that TLR3 has the protective role against WNV replication in brain and lethal encephalitis, as was demonstrated on the model of WNV-infected mice [116], others showed that the activation of TLR3 signaling is essential for WNV entry into the brain and thus predisposes the development of lethal encephalitis and permeability of the blood–brain barrier [117,122]. Furthermore, there are conflicting data, as one study did not observe an inhibitory effect of NS1 of several mosquito-borne flaviviruses including WNV, DENV, and YFV on TLR3 signaling, at least in HeLa and HEK293 cells [166] (Table 1). Crook and others authors of a later study, which demonstrated TLR3 inhibition by WNV NS1, suggested several reasons for such a discrepancy, such as different cell lines used for studies and differences in the expression levels of the studied signaling pathways components [164]. Thus, data on TLR3 signaling on WNV infection and on the role of NS1 in it are conflicting and demonstrate the need for more studies.

Several HCV nonstructural proteins are also involved in TLR signaling (Table 1). This is HCV NS3, which together with the core protein, is a specific TLR2 ligand, activates TLR2 signaling [167,168,169,170], and is also able to disrupt TLR3/4 signaling by cleaving TLR3 adaptor TRIF [171] (Table 1). This effect can also be reinforced by HCV NS4B, which is also shown to promote cleavage of TRIF, but by another mechanism including activation of caspase-8 which in its turn cleaves TRIF [172] (Table 1). HCV NS5A can also inhibit MyD88-dependent TLR signaling pathways by interacting with MyD88 [173] (Table 1). NS5A of the virus from another genus of *Flaviviridae-Pestivirus*, had a similar effect on MyD88, by interacting with it and thus decreasing MyD88 levels and MyD88-mediated TLR4 signaling in macrophages of BVDV-infected calves [174] (Table 1). An opposite effect on MyD88 was found for NS5A of CSFV (also from *Pestivirus* genus), which increased the activation of MyD88 and subsequently, elevated IFN-α levels in porcine monocyte-derived macrophages [175] (Table 1). At the same time, CSFV NS4B inhibited TLR3 signaling by an unknown mechanism [175]. Interestingly, HCV NS5A has also been noticed in two conflicting mechanisms: activating the TLR4 promoter, thus upregulating TLR4 expression in the model of B cells and hepatocytes [176]; and at the same time, in another study, reducing TLR4 expression in HepG2 hepatocytes and Huh-7 by an unknown mechanism [177] (Table 1). Taken together, HCV nonstructural proteins provide the activation of TLR2 and inhibition of TLR3 and of MyD88-dependent TLR signaling by different mechanisms. Data on TLR4 regulation by HCV NS5A are contradictory.

**Table 1 viruses-14-01808-t001:** *Flaviviridae* nonstructural proteins’ interference with TLR signaling.

Mediated Component of Signaling Pathway	Virus	Flaviviral Nonstructural Protein	Mechanism of Protein Interference	References
TLR4	DENV	NS1	DENV-infected patients with hemorrhagic fever have lower TLR4 expression in monocytes and reduced response to TLR4 stimulation by LPS in PBMC, higher serum levels of NS1, and lower levels of nitric oxide and TNF-α during the acute febrile phase of the disease, compared to patients with milder dengue fever.	[143]
			Secreted NS1 activates murine macrophages and PBMC through TLR4 signaling, elevating mRNA levels of TNF-α, IL-6, IFN-β, IL-1β, and IL-12, thus behaving as a viral toxin analogous to LPS. NS1 colocalizes with TLR4 in PBMC. The use of TLR4 antagonist protects mice from vascular leak in a model of infection.	[136]
			DENV NS1, but not WNV NS1, triggers vascular leak in wt C57BL/6 mice, independently on the TLR4 or TNF-α signaling and on the production of TNF-α, IL-6, and IL-8 by human dermal endothelial cells, but rather dependent on the effect of NS1 disruption of endothelial glycocalyx components.	[139]
			NS1 binds to platelets TLR4 and that triggers platelet aggregation and enhances platelet adhesion to endothelial cells and phagocytosis by macrophages, which may contribute to thrombocytopenia and hemorrhage during DENV.	[140]
			NS1 stimulation of platelets induces translocation of α-granules and release of stored factors such as RANTES/CCL5 and MIF. Both NS1 and DENV induce pro-IL-1β synthesis, but only DENV induces caspase-1 activation and subsequent IL-1β release by platelets. Platelet activation by NS1 partially depends on TLR4 but not TLR2/6 and NS1 interacts with TLR4. DENV infection of platelets is abortive, with no release of viral progeny, but NS1, expressed and released by platelets, activates infected platelets through an autocrine loop.	[141]
			Secreted NS1 protein of the four DENV serotypes increases endocan mRNA expression in endothelial cells by TLR4-dependent mechanism. Endocan is highly specific biomarker of endothelial cell activation, and its high levels are associated with lymphopenia and thrombocytopenia in DENV-infected patients and are correlated with the serum positivity for NS1 antigen.	[142]
			NS1 activates TLR4 but not TLR2/6 signaling in mouse macrophages and PBMC. Commercial *E.coli* derived NS1 used in the other study exhibits wrong protein folding and is contaminated by several microbial TLR ligands.	[135]
TLR2, TLR6			NS1 activates TLR2 and TLR6 in PBMC leading to increased production of IL-6 and TNF-α.	[144]
TLR3 (and TLR4/7)	WNV	NS1	NS1 inhibits cytokine production mediated by TLR3 in HeLa cells and by TLR4/TLR7 in bone marrow-derived mouse macrophages (BMDM) and bone marrow-derived myeloid dendritic cells (BMDC). Secreted NS1 binds predominantly to macrophages and DC in draining lymph nodes of mice and is capable of inhibiting TLR-induced responses in vivo (early IL-6 and IFN-β responses). Individual amino acid changes P320S and M333V in NS1 remove inhibition of TLR3 signaling in HeLa infected with WNV, attenuating viral replication.	[163,164,165]
TLR3	WNV DENV YFV	NS1	No specific interaction between TLR3 and NS1 of all tested viruses and no downregulation of TLR3 in HEK293 expressing NS1 or infected with WNV, DENV, or YFV or HeLa expressing NS1 is found.	[166]
TLR2 (and TLR1, TLR6 as coreceptors)	HCV	NS3	NS3 (and the core protein) is specific TLR2 ligand in macrophages, microglia, corneal epithelial cells, with TLR1 and TLR6 acting as coreceptors; activating TLR2 signaling; NF-κB signaling and inducing IL-10, IL-6, IL-8, IL-1β, and TNF-α secretion and nitric oxide production. TLR2-mediated cell activation is dependent on the conformation of NS3.	[167,168,169,170]
TRIF (TLR3/4)		NS3/4A	NS3/4A cleaves TRIF thus disrupting TLR3 (and TLR4) signaling in HeLa cells. No evidence for in vivo NS3/4A-mediated proteolysis of TLR3, TBK1, or IKKε is found.	[171]
TRIF (TLR3)		NS4B	NS4B activates caspase-8, which promotes cleavage and downregulation of TRIF and inhibition of TLR3-mediated IFN signaling in Huh7 cells.	[172]
MyD88		NS5A	NS5A interacts with MyD88 and inhibits IRAK recruitment to MyD88 in mouse macrophages.	[173]
TLR4 gene promoter			NS5A activates promoter of TLR4 gene and thus upregulates TLR4 expression in B cells (Raji cells) and hepatocytes, which mediates increased secretion of IFN-β and IL-6.	[176]
TLR4			NS5A reduces TLR4 expression in HepG2 hepatocytes and Huh-7 cells; inhibits mRNA expression of CD14, MD-2, MyD88, IRF3, and NF-κB2; and disrupts TLR4-mediated apoptosis by diminishing poly(ADP) polymerase cleavage, the activation of caspases 3, 7, 8, and 9 and by increasing the expression of anti-apoptotic molecules Bcl-2 and c-FLIP. No NS5A interaction with TLR4 is detected.	[177]
TLR4	BVDV	NS5A	Macrophages of BVDV-infected calves display decreased levels of TNF-α, IL-1β, and IL-6 in response to LPS; decreased TLR4 signaling; and reduction of MyD88 expression, likely due to NS5A interaction with MyD88 in macrophages. By contrast, monocytes exhibit elevated levels of TNF-α, IL-1β, and IL-6 in response to LPS.	[174]
TLR3	CSFV	NS4B, NS5A	NS4B inhibits activation of the TLR3 signaling pathway in porcine monocyte-derived macrophages (pMDM) and secretion of IL-6 and IFN-β. NS5A significantly increases the activation of MyD88 and IRF7 which results in the consequent synthesis of IFN-α in pMDMs.	[175]

## 3. RLRs and *Flaviviridae* Nonstructural Proteins

### 3.1. RLR Signaling

RLRs (Retinoic-acid inducible gene I (RIG-I) like receptors) are cytosolic PRRs [178] expressed in most cell types, which sense viral RNA or cellular RNA, which is unusual, mislocalized, or misprocessed [179]. The group includes RIG-I [180,181], MDA5 (melanoma differentiation-associated 5) [181,182], and LGP2 (laboratory of genetics and physiology 2) [183]. RLRs recognize cytoplasmic viral RNA via RNA binding motifs; the RLR signaling domain interacts with downstream adapter molecules and signaling cascades are activated which leads to the production of type I IFNs and proinflammatory cytokines and chemokines [75,184]. Furthermore, 5′-triphosphate and diphosphate of dsRNA are crucial for the recognition of viral RNA by RIG-I, whereas 5′-phosphate of dsRNA is dispensable for MDA5 activation [185,186,187]. All three above-mentioned RLRs have a central helicase domain, which enhances affinity to dsRNA through conformational change and ATP hydrolysis [188,189]. The helicase domain is followed by the CTD (C-terminal) domain, also known as the repressor domain RD, which recognizes dsRNA and ssRNA with three phosphates at 5′-end [189]. RIG-I and MDA5 additionally have two N-terminal CARDs (caspase activation and recruitment domains), responsible for the activation of downstream IRF3 and NF-κB transcription factors [78,188]. LGP2 lacks a CARD domain and thus is not able to autonomously transduce signaling. It plays a regulatory role in the activation of RIG-I and MDA5 [179] and is essential in antiviral responses [190].

RIG-I/MDA5 signaling is dependent on the adapter protein MAVS (mitochondrial antiviral-signaling protein), which is also variously known as the IFN-β promoter stimulator-1 (IPS-1), VISA (virus induced signaling adaptor protein), or CARDIF (CARD adaptor inducing IFN-β). MAVS is located on the outer membranes of mitochondria, peroxisomes, and in the mitochondrial-associated endoplasmic reticulum membrane (MAM) [191], and contains a CARD domain and a C-terminal transmembrane domain (TM) [179]. MAVS signaling is necessary for type I IFNs production in most cell types [192], and may also contribute to the recruitment of plasmacytoid DC [193]. When CTD of RIG-I and MDA5 binds to the viral ligand, RIG-I and MDA5 undergo a conformational change, and their CARD domains interact with the CARD domain of MAVS. MAVS is anchored with its TM into mitochondria, peroxisomes, and MAMs, which results in a downstream activation of TRAF3. TRAF3 catalyzes its own K63 poly-ubiquitination, thus recruiting TBK1 and IKKε, which in turn activate IRF3 and IRF7 [184,194]. After that, IRF3/7 homodimers and/or heterodimers translocate to the nucleus and bind to ISREs, which leads to the expression of type I IFNs [78,191] and subsequently activates the expression of ISGs. NF-κB is also activated by the RLR signaling pathway. The trigger for its activation is the interaction of MAVS with FADD (FAS-associated death domain-containing protein) and RIP1 via its non-CARD region [195]. The formation of the FADD/RIP1 complex is followed by the recruitment of the IKK complex [78,191], which mediates NF-κB translocation to the nucleus [78,84,195].

Sensing and signaling activities of RIG-I and MDA5 should be under tight control in order to maintain immune homeostasis in a normal physiological state and to provide an effective antiviral defense during infections. The short descriptions of only some of these control mechanisms, for which flavivirus nonstructural proteins’ interference has been demonstrated, are mentioned below.

The relocation of RIG-I and MDA5 from the sites of RNA recognition to mitochondria and MAMs is an essential event in RLR signaling and is mediated by the members of the 14-3-3 protein family. The 14-3-3ε protein forms a complex with RIG-I and TRIM25 (Tripartite motif-containing protein 25), stabilizes their interaction, and mediates their translocation to MAVS on MAMs [196]. Similarly, the 14-3-3η isoform behaves as a chaperone for MDA5, providing its aggregation and redistribution to mitochondria [197]. Several E3 ubiquitin ligases modify RIG-I, including the above-mentioned TRIM25 [198] and Riplet [199,200], which are essential for RIG-I signaling activation. Upon binding to dsRNA, RIG-I undergoes conformational changes [201,202,203], which give access of Riplet to RIG-I [204,205]. Riplet provides K63-polyubiquitination of RIG-I CTD, which is a repressor domain, and thus releases autorepression of RIG-I CARDs [200,206,207]. Recent studies also showed that Riplet ubiquitinates not only RIG-I CTD but also RIG-I CARDs [208,209], thus creating multiple ubiquitinated sites on RIG-I and promoting RIG-I activation. Riplet binding to RIG-I further promotes TRIM25 binding to RIG-I and TRIM25-dependent K63-polyubiquitination of RIG-I CARDs [198]. This event allows the binding of RIG-I CTD through its polyubiquitin chain to the ubiquitin-binding domain of the NEMO protein, forming the complex with TBK1 and IKKε [206]. Thus, the RIG-I-NEMO-TBK1-IKKε complex is formed, relocates to mitochondria, binds to MAVS, and oligomerizes leading to IRF3 activation [210]. The stability of TRIM25 and its interaction with RIG-I are negatively regulated by LUBAC (linear ubiquitin chain assembly complex), which conjugates K48-linked polyubiquitin chains to TRIM25, thus targeting TRIM25 to proteasome degradation [211]. The NEMO protein, a member of the IKK complex, is also a target of LUBAC. LUBAC binds to the IKK complex, and can modify NEMO by linear polyubiquitylation, which promotes efficient activation of the NF-κB response to TNF-α [212,213].

### 3.2. Flaviviridae Nonstructural Proteins’ Interference with RLR Signaling

RIG-I can sense plenty of *Flaviviridae* species, including YFV, TBEV, WNV, JEV, DENV, CSFV, and ZIKV [214,215,216,217,218,219,220,221]. Recognition by MDA5 also plays an important role in responding to various *Flaviviridae* infections, including WNV, DENV, HCV, CSFV, and ZIKV [219,220,221,222,223,224].

The suppression of the initial sensors of the RLR pathway, MDA5 and RIG-I, is demonstrated for several flaviviral nonstructural proteins (Table 2). Direct interaction with RIG-I is detected for WNV NS1 [225], ZIKV NS5 [226], and with MDA5—for NS4B of BVDV [227] (Table 2). The mechanisms of inhibiting RIG-I by ZIKV NS5 and WNV NS1 are the repression of K63-linked polyubiquitination of RIG-I [226]. These proteins thus inhibit RLR downstream signaling, NF-κB, and type I IFN responses. HCV NS5A [228] and ZIKV NS2A and NS4A [229] are also shown to suppress RIG-I and MDA5 signaling, though no evidence of their direct interaction with RLRs is found [228,229] (Table 2). Interestingly, HCV NS5B, on the contrary to the HCV NS5A protein which inhibited RIG-I and MDA5, creates the conditions of enhancing MAVS-dependent RLR signaling by its RdRp activity, and using cellular RNA as a template, catalyzes the production of small RNA species, which further could be sensed by RLRs [230] (Table 2). The same mechanism of upregulating RIG-I and MDA5 has also been found for TBEV NS5 [231] (Table 2).

Flaviviral nonstructural proteins, predominantly proteases, can inhibit RLR signaling also through its adaptor MAVS, by different mechanisms (Table 2). ZIKV NS3 catalyzes MAVS K48-linked ubiquitination, which leads to MAVS degradation [230] (Table 2). ZIKV NS4A, DENV NS4A, DTMUV NS1, and CSFV NS4B bind to MAVS, thus blocking any of its further interactions [232,233,234,235,236] (Table 2). HCV protease NS3/4A directly cleaves MAVS [237,238,239,240], notably, both peroxisomal and mitochondrial forms, which in both cases lead to the suppression of IFN activation [241] (Table 2). The clinical effect of this cleavage is demonstrated in HCV chronic patients. MAVS cleavage is detected in the samples of liver biopsies of these patients, and is more extensive in patients with a high HCV viral load, resulting in reduced activation of the endogenous IFN system in the liver [242]. All these processes block RLR signaling, but one of the flaviviral proteins, CSFV NS4A, has an effect of enhancing the MAVS pathway [243] (Table 2). Intriguingly, that CSFV NS4B protein vice versa inhibits MAVS and its downstream signaling [236] (Table 2).

RLR signaling may also be inhibited by the effects on scaffold 14-3-3 proteins, which normally help to relocate RIG-I and MDA5 to mitochondria (Table 2). ZIKV NS3 binds to and sequesters both 14-3-3ε and 14-3-3η proteins [244], thus blocking translocation of both RIG-I and MDA5, and DENV NS3 binds to 14-3-3ε and prevents translocation of RIG-I to MAVS [245] (Table 2). HCV NS3/4A is also involved in the cleavage of Riplet. The cleavage of Riplet by NS3/4A abrogates Riplet-mediated K63-linked polyubiquitination of RIG-I which is essential for the release of RIG-I autorepression and for subsequent RIG-I association with TRIM25 ubiquitin ligase and TBK1 protein kinase [206] (Table 2). Thus, HCV exhibits the redundancy in the inhibition of RLR signaling and inhibits antiviral responses by several means, including the above-mentioned cleavage of MAVS by NS3/4A and suppression of RIG-I and MDA5 by HCV NS5A. This may be the reason for the fact that even in the presence of the MAVS mutant, resistant to the cleavage by NS3/4A, RIG-I signaling is still reduced by HCV NS3/4A [238] (Table 2). The cleavage of another HCV NS3/4A substrate, TRIF, discussed above in the TLR signaling section, also contributes to the inhibition of TLR3 signaling [171], adding to the HCV redundancy in evading innate immunity.

ZIKV strategies of RLR inhibiting are also redundant, as they include suppression of MDA5, RIG-I, MAVS, 14-3-3ε, and 14-3-3η by its NS2A, NS3, NS4A, and NS5 proteins [226,229,232,233,244,246] (Table 2). However, there are some conflicting results on the role of ZIKV proteins, as Ngueyen et al. specified that they did not detect the effect of ZIKV NS3 or NS2B3 on MAVS levels in HEK293T transfected cells and on subsequent IRF3-mediated activation of IFN responses [229]; meanwhile, Li W. et al. pointed out the NS3 effect on MAVS [246] (Table 2). Ngueyen et al. attributed such discrepancies to the different cell lines, viral strains, and tags used for the studies [229]. Furthermore, in the study by Lundberg R. et al. in the model of HEK293 cells transiently transfected with RIG-I, luciferase gene under IFN promoter and plasmids encoding ZIKV proteins, only NS5 was shown to interfere with RIG-I signaling [247], which also contradicts the above-mentioned studies.

**Table 2 viruses-14-01808-t002:** *Flaviviridae* nonstructural proteins’ interference with RLR signaling.

Mediated Component of Signaling Pathway	Virus	Flaviviral Nonstructural Protein	Mechanism of Protein Interference	References
RIG-I, MDA5	WNV	NS1	NS1 interacts with RIG-I and MDA5 and targets them for proteasome degradation, and abrogates the K63-linked polyubiquitination of RIG-I, thereby inhibiting the activation of downstream RLR signaling pathway and of the IFN-β promoter in HEK293T cells.	[225]
RIG-I, MDA5 (and TBK1, IKKε, IRF3)	ZIKV	NS2A, NS4A	NS2A and NS4A suppress several components of RLR signaling in HEK293T cells, including RIG-I, MDA5, TBK1, IKKε, and IRF3, and thus, inhibit activation of IFN-β promoter.	[229]
RIG-I		NS5	NS5 binds to the CARD domain of RIG-I and subsequently represses K63-linked polyubiquitination of it, attenuating the phosphorylation and nuclear translocation of IRF3, thus inhibiting IFN-β response, as shown in studies in A549, HEK293T, and HeLa cells. The NS5 conservative site D146 located in the methyltransferase domain is critical for RIG-I signaling suppression, though methyltransferase activity does not contribute to the suppression.	[226]
MAVS		NS3	NS3 interacts with MAVS and degrades it by catalyzing its K48-linked ubiquitination, negatively regulating IFN-β response in SH-SY5Y and HEK293T cells.	[246]
MAVS		NS4A	NS4A specifically binds the N-terminal CARD domain and C-terminal transmembrane domain of MAVS, blocking its interaction with RIG-I and MDA5, and impairing type I IFN induction, as shown in human trophoblasts, fibroblasts, HEK293T, and HeLa cells. NS4A does not interfere with TLR-mediated immune responses.	[232,233]
14-3-3ε, 14-3-3η		NS3	NS3 binds to and sequesters the scaffold proteins 14-3-3ε and 14-3-3η, and thus antagonizes RIG-I and MDA5 anti-viral responses in astrocytes and HEK293T cells. Binding to 14-3-3 is mediated by a negatively charged motif in NS3 conservative for ZIKV strains of African and Asian lineages and a similar motif is found in DENV and WNV.	[244]
RIG-I, MDA5	HCV	NS5A	Region in domain 2 of NS5A protein plays an important role in the suppression of RIG-I and MDA5-dependent IFN response elicited by HCV, which is demonstrated in Huh7 cells and in vivo in a mouse model and may favor replication of HCV in the course of infection.	[228]
RIG-I, MDA5, MAVS		NS5B	NS5B is shown to catalyze the production of small RNA species by using cellular RNA as a template, which activates MAVS-dependent RLR signaling and thus, NF-κB, type I IFNs, and IL-6 secretion in human hepatocytes and mouse liver, which result in liver damage.	[230]
MAVS		NS3/4A	In the model of Huh7, Huh8, and HEK293 cells, NS3/4A is shown to bind to MAVS in the mitochondrial membrane and to cleave it at Cys-508, dislocating N-terminal fragment of MAVS from mitochondria and disrupting the colocalization of IKKε with MAVS on mitochondrial membrane. This disrupts RIG-I and MAVS interaction and finally the activation of NF-κB and IRF3 and subsequent IFN-β promoter activation. Mutation of Cys-508 to Ala in MAVS retains normal mitochondrial localization of MAVS and IKKε in the presence of NS3/4A. Subcellular redistribution of MAVS and its cleavage is demonstrated in liver tissues of chronically infected patients.	[237,238,239,240]
			Both peroxisomal and mitochondrial forms of MAVS are efficiently cleaved by NS3 in hepatocytes, and in both cases, this leads to suppression of IFN response activation.	[241]
Riplet		NS3/4A	NS3/4A cleaves Riplet and thus abrogates Riplet-mediated K63-linked polyubiquitination of RIG-I which is essential for the release of RIG-I autorepression and for subsequent RIG-I association with TRIM25 ubiquitin ligase and TBK1 protein kinase. On a model of HCV-infected Huh7 cells, endogenous Riplet is shown to be essential for antiviral response against HCV infection.	[206]
			NS4A Y16 residue located in the protein transmembrane domain regulates a noncanonical Riplet-TBK1-IRF3-dependent, but a RIG-I-MAVS-independent signaling pathway that limits HCV infection in Huh7 cells.	[248]
MAVS	DENV	NS4A	NS4A is associated with the N-terminal CARD-like domain and the C-terminal transmembrane domain of MAVS, preventing the binding of MAVS to RIG-I, repressing RIG-I induced IRF3 activation, and, consequently, IFN production as shown in HEK293T and HeLa cells.	[234]
14-3-3ɛ		NS3	NS3 binds to 14-3-3ɛ and prevents translocation of RIG-I to MAVS and thus blocks antiviral signaling in HEK293T and Huh7 cells. A highly conserved phosphomimetic RxEP motif in NS3 is essential for the binding to 14-3-3ɛ. A recombinant mutant DENV is not able to bind to 14-3-3ɛ and shows impairment in antagonism of RIG-I, eliciting a markedly augmented innate immune response and enhanced T cell activation.	[245]
MDA5	BVDV	NS4B	NS4B reduces mRNA levels of MDA5 and directly interacts with N-terminal CARDs of MDA5, inhibiting IFN-β promoter activity in HEK293T cells.	[227]
MAVS	DTMUV	NS1	Among all DTMUV proteins, tested in the assays with HEK293 cells, only NS1 disrupts association of MAVS with RIG-I or MDA5, by interaction with C-terminal domain of MAVS, thereby suppressing virus-triggered IFN-β expression.	[235]
MAVS	CSFV	NS4A	NS4A enhances MAVS pathway and thus induces IL-8 production in swine umbilical vein endothelial cells.	[243]
		NS4B	NS4B interacts with MAVS, not changing its expression level, but inhibiting MAVS-mediated NF-κB activation and IRF3 expression. Inhibition of IL-8 expression by NS4B is demonstrated in porcine alveolar macrophages.	[236]
RIG-I, MDA5	TBEV	NS5	NS5 upregulates RIG-I and MDA5 in human glioblastoma cells and primary astrocytes, probably by generating dsRNA molecules.	[231]

## 4. *Flaviviridae* Nonstructural Proteins’ Interference with NF-κB and IFN Signaling

Activation of the NF-κB pathway and of the initial stages of IFN response are the processes downstream of TLR and RLR signaling. The detailed mechanisms of NF-κB pathway activation and of the initial stages of IFN response have been discussed above, in TLR signaling and RLR signaling sections, and the ways of interference with these pathways by flavivirus nonstructural proteins are discussed below (Table 3 and Table 4).

### 4.1. NF-κB Signaling

Viruses including *Flaviviridae* have evolved strategies to target the NF-κB pathway, which may have several outcomes. The most widespread strategy exploited by a large number of viruses is the inhibition of NF-κB which dampens proinflammatory and immune responses of the host [249,250,251]. Furthermore, by activating NF-κB, viruses may exploit its apoptotic functions, which helps to increase viral spread via phagocytic myeloid cells, as demonstrated for DENV, Sindbis virus, and reoviruses [252]. Activation of NF-κB by the other viruses, including HIV-1, HTLV-1, HBV, HCV, RSV, rotaviruses, and influenza viruses results in the same outcome promoting viral replication but exploits vice versa antiapoptotic functions of NF-κB, which help to inhibit virus-induced apoptosis [252]. The activation of NF-κB by oncogenic viruses such as EBV, HTLV-1, and herpesviruses facilitates oncogenic transformation of infected cells [253,254] and also aids in maintaining viral latency [252]. Interestingly, this strategy may become reversed as some of these viruses tend to suppress NF-κB signaling during the lytic cycle [254]. Thus, targeting NF-κB pathway signaling by viruses may contribute to both enhanced viral replication and modulation of host immune responses.

HCV NS2 and NS5B proteins are shown to upregulate NF-κB activation [255,256] (Table 3). The mechanism of it for NS5B is suggested to be the same as the activation of MAVS RLR signaling induced by NS5B described above [230], which is the synthesis of RNAs by NS5B possessing RNA polymerase activity, from cellular RNA templates and subsequent sensing of these molecules by lymphotoxin β receptor and the production of lymphotoxin β [256] (Table 3). Interestingly, HCV proteins also have an opposite inhibitory effect on NF-κB signaling, which is implemented by different mechanisms. HCV NS3 and NS5B inhibit TNF-α-mediated NF-κB activation, and NS3 is shown to directly bind to LUBAC, thus inhibiting the formation of the NEMO-LUBAC complex, the LUBAC-mediated linear ubiquitylation of NEMO, and the subsequent activation of NF-κB [257] (Table 3). HCV NS5B is also shown to interact with IKKα and IKKβ and thus to inhibit subsequent IkBa degradation [258] (Table 3). Lastly, concerning HCV proteins, NS5A negatively regulates TNF-α induced NF-κB activation by interaction with TRAF2. TRAF2 is one of the proteins that helps to recruit TRADD to TNFR1 and is required for NF-κB activation [259] (Table 3).

DENV NS2B3, contrary to HCV NS3, promotes activation of NF-κB (Table 3), which results in endothelial cell death as demonstrated in the model of dengue hemorrhage in mice. NS3 cleaves cellular IκBα and IκBβ, activates the IKK complex, and thus activates NF-κB [260] (Table 3). Interestingly, the NS5A protein of another virus, CSFV, has an opposite effect on IkBα, suppressing its degradation and thus, NF-κB nuclear translocation and subsequent production of IL-1β, IL-6, and TNF-α in porcine alveolar macrophages [261] (Table 3). Coming back to DENV, DENV polymerase NS5 can bind to NF-κB binding sites on the RANTES promoter, thus activating the production of RANTES, which is one of the main proinflammatory chemokines [262] (Table 3).

In addition, two different mechanisms of interaction with TRAF6 are found for flaviviral nonstructural proteins but both promote viral replication (Table 3). CSFV NS3 degrades TRAF6 and thus inhibits TRAF6-dependent activation of NF-κB, which in the absence of NS3 impairs CSFV replication [263] (Table 3). NS3 of Langat virus, which is a TBEV-related virus, also interacts with TRAF6, and it is shown to promote the accumulation of the mature processed NS3 and playing a proviral role. This effect has not been found for mosquito-borne flaviviruses, and two putative tumor necrosis factor receptor-associated factor 6 (TRAF6)-binding motifs (TBMs) within the protease domain of Langat NS3 are detected only in tick-borne, but not mosquito-borne flaviviruses [264] (Table 3).

The unique mechanism of NF-κB inhibition is described for JEV NS5. NS5 interacts with the proteins of nuclear transport which help to translocate IRF3 and p65 (NF-κB subunit) to the nucleus, thus inhibiting downstream NF-κB and IFN signaling (Table 3 and Table 4 [265]).

**Table 3 viruses-14-01808-t003:** *Flaviviridae* nonstructural proteins’ interference with NF-κB signaling.

Mediated Component of Signaling Pathway	Virus	Flaviviral Nonstructural Protein	Mechanism of Protein Interference	References
NF-κB	HCV	NS2	NS2 upregulates NF-κB activation, and thus, CXCL8 (IL-8) transcription in HepG2 cells.	[255]
NF-κB (and LTβ)		NS5B	NS5B activates lymphotoxin signaling and thus upregulates lymphotoxin β production (LTβ). This upregulates its downstream targets NF-κB and CXCL10 in HCV-related tumors and in Huh7 cells. The suggested mechanism of upregulation is RNA synthesis provided by RNA polymerase activity of NS5B, most probably, from cellular RNA templates. Subsequent sensing of these RNA molecules by LTβ receptor activates the production of LTβ.	[256]
LUBAC		NS3 and NS5B	NS3 directly interacts with LUBAC thus competing with NEMO for binding to LUBAC, and inhibits the LUBAC-mediated linear ubiquitylation of NEMO and the subsequent activation of NF-κB in Huh7 cells. Thus, NS3 inhibits the TNF-mediated activation of NF-κB. NS5B is shown to have the same effect on TNF-mediated activation of NF-κB.	[257]
TRAF2		NS5A	NS5A interacts with TRAF2 and is required for the activation of NF-κB in Cos7 and HEK293T cells. Interaction of NS5A with TRAF2 leads to the negative regulation of TNF-α-induced NF-κB activation.	[259]
IKKα, IKKβ		NS5B	NS5B interacts with IKKα and IKKβ (but not IKKγ), and thus inhibits TNF-mediated IKK activation and subsequent IκBα degradation. NS5B inhibits TNF-, TRAF2-, and IKK-induced NF-κB activation. At the same time, NS5B protein synergistically activates TNF-α-mediated JNK activity in HEK293 and hepatic cells.	[258]
IκBα, IκBβ	DENV	NS3 (NS2B3)	NS3 interacts with cellular IκBα and IκBβ and cleaves them and induces IKK activation in endothelial cells. This activates NF-κB, which results in endothelial cell death (demonstrated on the mouse model of dengue hemorrhage). NS3 enzymatic activity is crucial for this effect.	[260]
NF-κB		NS5	NS5 binds to NF-κB binding sites of the RANTES promoter, thereby activating RANTES production in HEK293 cells.	[262]
TRAF6	LGTV	NS3	NS3 interacts with TRAF6 (tumor necrosis factor receptor-associated factor 6) during TBEV infection, providing the accumulation of the mature processed protease and thus playing a proviral role in HEK293 cells and mouse embryonic fibroblasts. Two putative TRAF6-binding motifs within the protease domain not present in mosquito-borne flaviviruses, are essential for TRAF6 binding to NS3 and for its proviral role.	[264]
TRAF6	CSFV	NS3	The overexpression of TRAF6 has an inhibitory effect on CSFV replication, apparently, through TRAF6-dependent activation of NF-κB, which subsequently increases IFN-β and IL-6 expression in porcine alveolar macrophages. CSFV infection or sole expression of CSFV NS3 degrades TRAF6, thus contributing to persistent CSFV replication.	[263]
NF-κB		NS5A	NS5A suppresses IkBα degradation, NF-κB nuclear translocation and NF-κB activity, and inhibits IL-1β, IL-6 and TNF-α expression in porcine alveolar macrophages.	[261]
KPNA2, KPNA3, KPNA4, importin-alpha 4, and importin-alpha 3	JEV	NS5	NS5 interacts with the nuclear transport proteins KPNA2, KPNA3, KPNA4, importin-alpha 4, and importin-alpha 3, thus competitively blocking interaction of these molecules with their cargo molecules IRF3 and NF-κB subunit p65 and inhibiting their translocation to the nucleus, and finally, IFN-β expression, as shown in HeLa and HEK293T cells.	[265]

### 4.2. IFN Signaling

The NS5s of several flaviviruses, including TBEV, JEV, WNV, ZIKV, and DENV have been shown to activate RANTES expression [231] (Table 4). For TBEV, the mechanism of this effect is specified. NS5, as described in the RLR signaling section, aids in producing dsRNAs in the cells, and thus upregulates RIG-I and MDA5 signaling and also, the phosphorylation of IRF3, inducing its translocation to the nucleus and its binding to ISRE of the RANTES promoter [231] (Table 4).

Several ZIKV proteins, NS1, NS4B, and NS5 are shown to interact with TBK1 and thus antagonize the IFN response: NS1 and NS4B by blocking oligomerization of TBK1 [266] and NS5 by binding TBK1 ubiquitin-like domain and impairing the formation of the TBK1-TRAF6 complex [267] (Table 4). Additionally, ZIKV NS2B has been demonstrated to inhibit JAK-STAT signaling by degrading JAK1 and thus reducing virus-induced apoptotic cell death [266]. Furthermore, the cooperation of ZIKV NS1, NS2B3, and NS4 proteins blocks IFN-induced autophagy degradation of NS2B3 and thus enhances viral infection [266] (Table 4). To add to the mechanisms employed by ZIKV proteins, NS5 interacts with IKKε, inhibits its phosphorylation, and subsequently the phosphorylation of IRF3 [247] (Table 4). The same effect is demonstrated for DENV NS2B3 protease [268] (Table 4).

Inhibition of IFN signaling through interaction with TBK1 is also demonstrated for HCV NS3/4A (though Li K. et al. who published the paper the same year, did not find any evidence for in vivo NS3/4A-mediated proteolysis of TBK1 [171]), DENV NS2A, and NS4B and WNV NS4B [269,270] (Table 4). Interestingly, for DENV NS4A among the DENV serotypes that are tested (1, 2, and 4), the NS4A of only DENV1 uniquely inhibits TBK1, unlike NS4B, the inhibitory effect of which on TBK1 is shown for DENV1/2/4 serotypes [270]. Of note, infections caused by DENV1 are characterized by increase severity and DENV1 responses are immunodominant during tetravalent DENV1–4 vaccination [270].

**Table 4 viruses-14-01808-t004:** *Flaviviridae* nonstructural proteins’ interference with IFN signaling.

Mediated Component of Signaling Pathway	Virus	Flaviviral Nonstructural Protein	Mechanism of Protein Interference	References
IRF3	TBEV (JEV, WNV, ZIKV, DENV)	NS5	NS5 of TBEV, JEV, WNV, ZIKV, DENV activates RANTES expression in human glioblastoma cells (and also primary astrocytes for TBEV NS5). TBEV NS5 upregulates RIG-I and MDA5 and, due to its RdRp activity, upregulates the phosphorylation of IRF3 and induces its translocation to the nucleus and binding to ISRE of RANTES promoter.	[231]
IKKε	ZIKV	NS5	Among all ZIKV proteins, only NS5 interferes with RIG-I signaling pathway in HEK293 cells. NS5 interacts with IKKε, inhibits its phosphorylation and phosphorylation of IRF3. In addition, NS5 inhibits activation of IFN-λ1 promoter and of NF-κB.	[247]
TBK1 (andJAK1)		NS1, NS4B, NS2B3	NS1 and NS4B interact with TBK1 and block its oligomerization. NS2B3 degrades JAK1 and reduces virus-induced apoptotic cell death. This inhibits IFN response. Cooperation of NS1, NS4B, and NS2B3 enhances viral infection by blocking IFN-induced autophagy degradation of NS2B3 in HEK293T cells.	[266]
TBK1		NS5	NS5 of ZIKV MR766 strain interacts with the ubiquitin-like domain of TBK1 and results in impaired interaction of TBK1 and TRAF6, thus dampening TBK1 activation, IRF3 phosphorylation, and antagonizing IFN production in HEK293T cells.	[267]
TBK1	HCV	NS3/4A	NS3 interacts with TBK1 kinase, inhibiting its association with IRF3 and IRF3 activation, as shown in HEK293T and Huh7 cells.	[269]
IRF3, IKKε	DENV	NS2B3	NS2B3 interacts with IKKε, masking the kinase domain and preventing the phosphorylation and nuclear translocation of IRF3 in HEK293 cells.	[268]
TBK1	DENV, WNV	NS2A, NS4A, NS4B	DENV NS2A and NS4B inhibit RIG-I-, MDA5-, MAVS-, and TBK1/IKKε-directed IFN-β transcription but not IFN-β induction directed by STING or constitutively active IRF3 in HEK293T cells. NS2A and NS4B from DENV1/2/4, as well as WNV NS4B, commonly inhibit TBK1 phosphorylation and IFN-β induction.	[270]
KPNA2, KPNA3, KPNA4, importin-alpha 3, importin-alpha 4	JEV	NS5	NS5 interacts with the nuclear transport proteins KPNA2, KPNA3, KPNA4, importin-alpha 4, and importin-alpha3, thus competitively blocking interaction of these molecules with their cargo molecules IRF3 and NF-κB subunit p65 and inhibiting their translocation to the nucleus, as shown in HeLa and HEK293T cells.	[265]

## 5. cGAS-STING Signaling and *Flaviviridae* Nonstructural Proteins

### 5.1. Cytoplasmic DNA Sensors

Beyond cytoplasmic RNA sensors, cytoplasmic DNA sensors are known, such as cGAS (cyclic GMP-AMP synthetase, or cGAMP synthetase). cGAS is activated by binding to cytoplasmic dsDNA, and catalyzes the cyclization of AMP and GMP in the cytoplasm producing the noncanonical cyclic dinucleotide 2′3′-cGAMP, which serves as the second signal, binding to STING (stimulator of IFN genes) dimers [271]. This results in the translocation of STING from ER membrane to Golgi bodies, triggers the activation of TBK1 and IKK complexes, and thus, the production of type I IFNs and NF-κB signaling [271,272]. DNA viruses, including adenoviruses, HSV-1, and MVA can lead to type I IFN production via the cGAS-STING pathway [272,273,274]. Interestingly, STING is identified as the central player in the crosstalk between DNA and RNA sensing. The example of such interaction is illustrated by JEV. JEV RNA is recognized by RIG-I, which then recruits STING to initiate a downstream cascade leading to the antiviral response [275]. In this study and other studies, STING appeared to form a complex with RIG-I and MAVS, which was stabilized upon virus infection [275,276,277].

### 5.2. Flaviviridae Nonstructural Proteins’ Interference with cGAS-STING Signaling

Interestingly, cGAS, initially known as a sensor of microbial DNA, was found playing an important role in limiting infections by RNA viruses. cGAS detects members of *Flaviviridae* and may limit their replication [278,279]. One study showed that DENV disturbs the mitochondrial membrane and thus leads to the leakage of mitochondrial DNA in the cytoplasm, where it is sensed by cGAS and activates STING [278,280]. However, a later study by Su CI et al. demonstrated that not endogenous mitochondrial DNA but rather exogenous reactivated viral DNA (i.e., in case of reactivation of latent infections by DNA viruses) could be sensed by cGAS during DENV infection [281]. ZIKV also promotes infection by antagonizing STING [282]. JEV can activate cGAS-STING signaling in the model of mouse embryonic fibroblasts infection [283].

DENV NS2B3 has been demonstrated to cleave human STING and thus to inhibit IRF3 activation and IFN response in several studies [281,284,285,286,287] (Table 5). Interestingly, neither murine, nor non-human primates STING can be cleaved by NS2B3 [284,287], and in DENV-infected murine cells, STING strongly restricts DENV replication [284] (Table 5). Different STING haplotypes have different sensitivity to DENV protease [281]. An additional mechanism used by DENV NS2B to evade cGAS-STING signaling is targeting of cGAS to lysosomal degradation [280] (Table 5). ZIKV NS1 also, through complex mechanism, promotes cleavage of cGAS [288] (Table 5), which decreases recognition of DNA during infection and inhibits type I IFN. Cleavage of human STING has also been demonstrated by NS2B3 of ZIKV, DENV, JEV, and WNV [246,282], and of duck STING—by DTMUV NS2B3 [289] (Table 5). DTMUV has a redundant mechanism of STING inhibition as DTMUV NS2A interacts with STING, disrupting the formation of STING-STING and STING-TBK1 complexes [290] (Table 5). Thus, STING cleavage seems to be a universal mechanism for viral proteases to evade innate immunity.

YFV and HCV NS4B protein have also been shown to interact with STING and thus to disrupt its interactions with downstream proteins [291,292,293,294] (Table 5). Interestingly, another HCV protein with protease activity, NS3/4A, is demonstrated to cleave MAVS, the component of RLR signaling [293]. This is an illustration of cooperation of flaviviral proteins where different viral proteases cleave different substrates from different stages of innate immune pathways, but promote a universal result which is antagonizing IFN responses [293].

**Table 5 viruses-14-01808-t005:** *Flaviviridae* nonstructural proteins’ interference with cGAS-STING signaling.

Mediated Component of Signaling Pathway	Virus	Flaviviral Nonstructural Protein	Mechanism of Protein Interference	References
STING	DENV	NS2B3	On models of A549, Huh7 cells, and primary human and mouse monocyte-derived dendritic cells, NS2B3 is demonstrated to interact with human STING cleaving it at LRR↓^96^G. The sensitivity of STING to DENV NS2B3 varies with different STING haplotypes. NS3 recruitment to NS2B and the formation of NS2B3 are facilitated by K27-linked polyubiquitination of NS3 protein, which enhances STING cleavage. Activation of IRF3 and type I IFN induction are thus blocked. The endoplasmic reticulum protein SCAP binds NS2B protein and thus inhibits ubiquitination of NS3 and hinders NS2B3 from binding to and cleaving STING. Murine and non-human primates STINGs are not cleaved by NS2B3.	[281,284,285,286,287]
cGAS		NS2B	NS2B targets cGAS for lysosomal degradation, which helps to avoid the detection of mitochondrial DNA by cGAS during infection and results in type I IFN response inhibition in primary human monocyte-derived dendritic cells.	[280]
STING	ZIKV, DENV, JEV, WNV	NS2B3	NS2B3 of DENV, ZIKV, JEV, WNV, but not of YFV cleaves human but not murine STING, as shown in a model of fibroblasts.	[282]
STING	ZIKV	NS3, NS2B3	NS2B3 interacts with STING (called MITA in this study), catalyzing its K48-linked ubiquitination, and thus degrades it, negatively regulating IFN-β response in SH-SY5Y and HEK293T cells. NS2B3 also impairs K63-linked polyubiquitination of STING, which is essential for IFN signaling cascades.	[246]
cGAS		NS1	NS1 recruits host deubiquitinase USP8 to cleave K11-linked polyubiquitin chains from caspase-1 at Lys134. This inhibits the proteasomal degradation of caspase-1 and stabilizes caspase-1, which promotes the cleavage of cGAS. The enhanced cleavage of cGAS leads to the decrease in recognition of mitochondrial DNA release, and thus inhibits type I IFN signaling, as shown for THP-1 cells, PBMC, and mouse brain tissues.	[288]
STING	HCV	NS4B	NS4B interacts with STING and disrupts its interaction with TBK1, which blocks type I and III IFN responses in HEK293 cells and hepatocytes. NS4B does not have any effect on the interaction of STING with MAVS.	[292]
STING		NS4B	NS4B (more specifically, transmembrane domain of NS4B of 2a JFH1 replicon, but not of 1b/Con1 replicon) suppresses STING accumulation during replication of HCV in human hepatoma cells, thus inhibiting IFN-β activation.	[294]
STING (and MAVS)		NS4B, NS3/4A	NS4B binds to STING blocking its interaction with MAVS and inhibiting IFN-β production in HEK293T and Huh7 cells. NS4B N-terminus containing STING homology domain is important in this interaction. NS3/4A does not suppress STING-induced IFN-β activation. At the same time, NS3/4A can cleave MAVS also inhibiting IFN-β activation, but not completely. Residual IFN-β activation is suppressed by NS4B, suggesting the cooperation of NS3/4A and NS4B in IFN-β antagonism.	[293]
STING	YFV	NS4B	NS4B inhibits STING activity in mouse embryonic fibroblasts, probably by direct association with it.	[291]
STING	DTMUV	NS2B3	NS2B3 cleaves duck STING due to its protease activity and thus inhibits RIG-I-, MDA5-, MAVS-, and STING-directed IFN-β transcription, but not TBK1- and IRF7-mediated induction of IFN-β in duck embryo fibroblasts. Binding of NS2B3 to STING is dependent on protease cofactor NS2B, but not NS3.	[289]
		NS2A	NS2A inhibits duck RIG-I-, MDA5-, MAVS-, STING-, and TBK1-induced IFN-β transcription, but not duck TBK1- and IRF7-mediated phases of IFN response in duck embryo fibroblasts. NS2A interacts with STING and thus disrupts the formation of STING-STING and STING-TBK1 complexes, reducing phosphorylation of TBK1. STING dimerization and phosphorylation are critical for its interaction with NS2A.	[290]

## 6. NLR signaling, Inflammasomes, and *Flaviviridae* Nonstructural Proteins

### 6.1. NOD-like Receptors and Inflammasome

The family of NOD-like receptors (NLRs) consists of more than 22 members, which are characterized by the presence of a central NOD or nucleotide binding (NBD) domain, also known as NACHT, flanked with a C-terminal LRR (leucine-rich repeat) domain, and a variable N-terminal interaction domain [295]. NLRs are located in the cytoplasm and are expressed in immune and non-immune cells, including lymphocytes, macrophages, DCs, and epithelial cells [296]. The NLRs recognize different PAMPs or DAMPs, and activate the NF-κB complex and the expression of proinflammatory and chemotactic cytokines [297].

The first NLRs found to detect bacterial components were NOD1 and NOD2, also called NLRC1 and NLRC2. They sense bacterial peptidoglycan, and this triggers the release of auto-inhibitory conformation, oligomerization of NOD1/2, and recruitment of the receptor-interacting protein kinases 2 (RIP2) via CARD–CARD interactions, and activation of NF-κB and MAPKs, which drive the expression of proinflammatory genes and antimicrobial responses [298,299]. Over recent years, more data on NOD ability to sense viral infections and individual viral proteins have accumulated.

The most widely studied NLRs, NLRP3, NLRP1, and NLRC4, represent a part of a multi-protein complex called inflammasome. Microbial PAMPs and DAMPS are sensed by these NLRs and trigger inflammasome activation [300]. Inflammasomes are composed of a sensor, an adaptor (present not in all types of inflammasomes), and an effector. Inflammasome sensors may be presented by NLRs or the other receptor types (AIM2, etc.). The most well-studied inflammasome has an NLRP3 sensor (NLR family pyrin domain-containing 3), and thus is usually called NLRP3 inflammasome [301]. NLRP3 is a cytoplasmic protein composed of three domains: N-terminal pyrin domain (PYD), the central nucleotide binding and oligomerization domain (NACHT), and the carboxyterminal leucine-rich repeat sequence (LRR) [302,303]. The NACHT domain is critical for activation and mediates ATP-dependent self-oligomerization, the LRR region senses PAMPs, and the PYD domain plays the role in homotypic protein–protein interaction [296,304].

The NLRP3 inflammasome complex also includes the adaptor protein ASC (an apoptosis associated speck-like protein containing a CARD) composed of two domains (PYD and CARD) and an effector protein pro-caspase-1 [305]. Activated by specific stimuli, an inflammasome complex assembles due to the interaction of PYD domains of NLRP3 and ASC. ASC in its turn recruits inactive pro-caspase-1 through CARD–CARD interactions [306,307]. The activation of NLRP3 also requires NEK7 (the mitotic NIMA-related kinase 7), which bridges adjacent subunits of NLRP3 and thus mediates inflammasome activation [308]. Inflammasome assembly leads to autoprocessing of pro-caspase-1 to its active form caspase-1. Active caspase-1 forms heterotetramer p20/p10 and cleaves pro-IL-1β/pro-IL-18 into mature IL-1β/IL-18 [309,310,311]. Additionally, caspase-1 cleaves pro-GSDMD (pro-gasdermin D), which leads to the formation of GSDMD pores, and the release of mature IL-1β and IL-18 takes place [312] and triggers a cascade of inflammatory reactions, and eventually leads to inflammatory cell death called pyroptosis [313,314].

The basal expression levels of NLRP3, pro-IL-1β, and pro-IL-18 are usually low, and a two-step process is required for priming and activation of the inflammasome [315,316]. The priming stimuli are PAMPs or DAMPs recognized by TLRs and cytokine receptors, such as TNFR, IL-1R (IL-1 receptor), or IFNAR. Priming leads to the activation of NF-κB and to the upregulation of NLRP3, pro-caspase-1, pro-IL-1β, and pro-IL-18 expression [317]. The activation step is also induced by numerous PAMPs and DAMPs, lysosomal rupture, or mitochondrial and ER stress [318,319], and promotes inflammasome assembly. The resulting secretion of IL-1β and IL-18 leads to the activation of immune cells and recruitment of neutrophils to the inflammatory site to aid the elimination of viruses, and also is important for the initiation of the adaptive immune response [320,321].

### 6.2. Flaviviridae Nonstructural Proteins’ Interference with NLR Signaling and Inflammasomes

Unsurprisingly, *Flaviviridae*, which cause inflammatory reactions, are capable of activating inflammasomes. This was demonstrated for several viruses, including JEV, DENV, WNV, HCV, and ZIKV, and in some cases, viral proteins are involved in these processes (Table 6), which are discussed below.

DENV infection leads to the increased expression of IL-1β in platelets and platelet-derived microparticles, and that can be a factor of increased vascular permeability. Mitochondrial ROS are considered as the factor of NLRP3 inflammasome activation [322]. Macrophages, which are the major target for DENV replication [323,324], are also shown to activate NLRP3 inflammasome in the presence of DENV, but this is characteristic only for the specific macrophage subset of inflammatory macrophages, and not resting macrophages [325]. In this study, the blockage of CLEC5A, which is a C-type lectin critical for the release of proinflammatory cytokines during DENV infection, inhibits NLRP3 inflammasome activation and pyroptosis, pointing at the possible role of CLEC5A in NLRP3 inflammasome regulation and in pathogenesis of DENV [325]. Interestingly, NLRP3 inflammasome together with FcγRIII, TLR3, and antiplatelet antibodies contribute to the induction of hemorrhage during secondary infection with DENV [326]. Over the last years, data on the role of several DENV proteins in triggering inflammasome activation have accumulated. The DENV envelope protein domain III (EDIII) induces IL-1β maturation in human macrophages, which is associated with an increase in ROS production and potassium efflux and is mediated by caspase-1 and NLRP3 inflammasome activation [327,328]. In addition, EDIII activates NLRP3 in endothelial cells which promotes their disfunction and is a factor of hemorrhage induction in mice [328]. A direct interaction with NLRP3 is demonstrated for DENV membrane protein M, which thus promotes the release of IL-1β and induces vascular leakage in mouse tissues [328]. Among DENV nonstructural proteins, NS2A and NS2B are demonstrated to increase NLRP3 inflammasome activation in endothelial cells, by acting as putative viroporins [329] (Table 6). The NS1 of DENV is also discussed as a possible activator of pyroptosis, as a treatment with NS1-induced platelet cell death, pyroptosis, and thrombocytopenia in mice, though not with as high levels as with EDIII [328]. NS1 is also a well-known viral factor of induction of DENV-associated inflammation [136]. Thus, more detailed studies of the molecular mechanisms of DENV induction of inflammation and of its proteins’ role in it are needed.

WNV induces NLRP3 inflammasome [330] and the critical role of ASC in this is demonstrated [331]. IL-1β signaling is synergetic with the type I IFN response and helps to suppress WNV replication in neurons, as IL-1βR-deficient mice exhibit increased susceptibility to WNV pathogenesis [330]. The combined treatment of neurons with IL-1β and IFN-β followed by WNV infection, leads to almost complete control of WNV and the induction of ISGs is detected; meanwhile, sole treatment either with IL-1β, or IFN-β provides only partial control over infection [330]. Thus, despite the known ability of WNV to antagonize type I IFN responses [225,282], this antagonism may be overcome by the synergy of IL-1β and IFN-β.

Several recent studies demonstrate that ZIKV infection activates or inhibits NLRP3 inflammasome and pyroptosis. Activation of inflammasome is demonstrated in both glial cells, which points at the possible important role of inflammasome activation in neuroinflammation caused by ZIKV [332], and in macrophages [333,334] which are considered the earliest blood cell type infected by ZIKV and dendritic cells. Moreover, the role of two ZIKV proteins is noticed in these processes (Table 6). ZIKV NS5 is shown to facilitate inflammasome assembly by direct association with NLRP3 through its RdRp domain [333,334], and probably also due to the generation of ROS induced by NS5 protein [333] (Table 6). The ZIKV NS1 protein facilitates inflammasome assembly by inhibiting the degradation of caspase-1 [288] (Table 6). This phenomenon is an excellent example of an antagonistic interaction between inflammasome and type I IFN signaling. NS1 activates inflammasomes and inflammatory responses. At the same time, it promotes cleavage of cGAS, and thus, the decrease in recognition of mitochondrial DNA, which leads to the inhibition of IFN signaling and determines high sensitivity to ZIKV. NLRP3-deficient mice demonstrate the enhanced type I IFN responses and protection from ZIKV [288]. Thus, inflammasome activation during ZIKV infection has deleterious effects on the protection from virus. This effect is opposed to WNV infection, where inflammasome activation and IL-1β secretion are synergetic with IFN-β response and provide control over viral replication [330].

Strangely, the other authors did not detect any effect of ZIKV NS1 and NS5 on the activation/inhibition of inflammasome, and ZIKV is shown to inhibit inflammasome activation by NS3 protease which induces NLRP3 degradation, probably, by cleaving it [335] (Table 6). The possible explanation of this discrepancy provided by the authors is different cell lines studied, as the activation is demonstrated in THP-1 and PBMC, and inhibition, in mouse macrophages and glial cells [335]. Additionally, they described different ZIKV lineages, Asian vs. African, which differ in their cytokine expression patterns [336]. Anyway, this example is an illustration of the lack of knowledge in the field of inflammasomes activation during *Flaviviridae* infections, and of the necessity of more studies. The induction of NLRP3- and caspase-1 dependent pyroptosis by ZIKV is demonstrated in human and murine macrophages [337]. Another study demonstrated on a model of human glioblastoma cells that ZIKV is also able to induce caspase-independent pyroptosis, provided by ZIKV NS2B3, which directly cleaves the GSDMD into N-terminal fragment [338] (Table 6). One study also suggested that ZIKV can activate AIM2 inflammasome in skin fibroblasts, as elevated mRNA levels of AIM2 are induced by ZIKV infection, but more studies upon this should be performed [339].

JEV induces IL-1β and IL-18 in microglia and astrocytes, and these cytokines may play a role in the neurotoxicity of JEV [340]. Later, on a murine model of JEV infection, it is demonstrated that production of reactive oxygen species and potassium efflux are critical for the maturation of NLRP3 inflammasome, activation of caspase-1, and subsequent secretion of IL-1β and IL-18 [341]. In addition, JEV could upregulate the expression of NOD1, leading to an enhanced neuroinflammatory response, and NOD1 activation is shown to play the role in the inflammatory response triggered by JEV infection [342]. Interestingly, JEV is found to develop the protection mechanism against host microRNAs [343] (Table 6). Certain miRNAs presented in the mouse brain suppress IL-1β secretion as well as JEV replication in neuroblastoma cells. JEV NS3 has been shown to be a potent miRNA suppressor, thus promoting neuroinflammation in the brain, and JEV replication [343] (Table 6).

HCV infection activates caspase-1 dependent IL-1β/IL18 secretion in macrophages including liver-resident macrophages [344,345]. The upregulation of pro-IL-1β/IL18 mRNA expression may occur through MyD88-mediated TLR7 signaling triggered by HCV RNA [345,346], and through the NF-κB signaling pathway [344] which can be triggered by HCV viroporin p7, shown to prime and to activate inflammasome [344]. In both cases, a potassium efflux seems to play a significant role in inflammasome activation [344,345]. Activation of inflammasome in HCV-infected hepatocytes which are the primary site of HCV infection is questionable, as several studies concluded that it is not activated [344,345,347]; meanwhile, in one recent study, the activation of inflammasome is detected in HCV-infected Huh7 hepatoma cells [348]. Furthermore, in the study by Chen W. et al. inflammasome activation in human monocytes and macrophages is not detected in cells treated with HCV virions, and is detected in macrophages only after transfection with HCV RNA [347]. This activation is NLRP3-mediated ROS-dependent [347]. HCV NS5B is shown to be the activator of NOD1, NLR, which is normally activated by bacterial components [349] (Table 6). Due to its RdRp activity, NS5B synthesizes RNAs from cellular templates, and these RNA molecules are sensed by NOD1, which demonstrates that NOD1 can also be the RNA ligand recognition receptor [349]. Interference with NOD1-mediated signaling significantly weakens the inflammatory response to dsRNA [349].

CSFV is also able to activate inflammasome and pyroptosis in macrophages and peripheral lymphoid organs [350,351], and CSFV viroporin p7 triggers IL-1β secretion in macrophages [351].

**Table 6 viruses-14-01808-t006:** *Flaviviridae* nonstructural proteins’ interference with NLR signaling and inflammasomes.

Mediated Component of Signaling Pathway	Virus	Flaviviral Nonstructural Protein	Mechanism of Protein Interference	References
NLRP3 inflammasome	DENV	NS2A, NS2B	Both DENV and its NS2A and NS2B proteins increase the NLRP3 inflammasome activation resulting in IL-1β secretion in endothelial cells. NS2A and NS2B behave as putative viroporins and solely stimulate the NLRP3 inflammasome complex in LPS-primed endothelial cells.	[329]
NLRP3	ZIKV	NS5	NS5 facilitates NLRP3 inflammasome complex assembly in differentiated THP-1 macrophages by its RdRp domain interacting with NLRP3 and by facilitating reactive oxygen species production, so NS5 is treated as a stress signal. When binding to NLRP3, NS5 forms a spherelike structure of NS5–NLRP3–ASC, in which NS5 locates inside, NLRP3 locates in the middle, and ASC distributes outside, as shown in HEK293T cells.	[333,334]
		NS3	ZIKV infection upregulates transcription of IL-1β and IL-6 by activating NF-κB signaling, but does not trigger NLRP3-dependent ASC oligomerization and secretion of active caspase-1 and IL-1β even after LPS priming and ATP stimulation of macrophages and glial cells. ZIKV NS3 overexpression leads to the degradation of NLRP3 (probably, by its cleavage). NS1 and NS5 proteins do not have any impact on NLRP3 inflammasome. ZIKV infection influences only NLRP3-dependent but not AIM2-dependent caspase-1 activation and IL-1β secretion.	[335]
Caspase-1, cGAS		NS1	ZIKV infection activates NLRP3 inflammasome, and NS1 protein facilitates inflammasome assembly, by recruiting host deubiquitinase USP8 to cleave K11-linked polyubiquitin chains from caspase-1 at Lys134. This inhibits the proteasomal degradation of caspase-1 and stabilizes caspase-1, which promotes both IL-1β release and the cleavage of cGAS. The enhanced cleavage of cGAS leads to the decrease in recognition of mitochondrial DNA release and thus inhibits type I IFN signaling, as shown for THP-1, PBMC, and mouse brain tissues.	[288]
GSDMD		NS2B3	On a model of human glioblastoma U87-MG cells, NS2B3 is shown to cleave the GSDMD into N-terminal fragment (1–249) leading to pyroptosis in a caspase-independent manner.	[338]
miR-466d-3p	JEV	NS3	On a model of neuroblastoma cells and mouse brains, NS3 is shown to suppress the expression of mature RNAs miR-466d-3p, which are found to regulate JEV-induced inflammation in the CNS. The helicase region of NS3 binds specifically to miRNA precursors and can lead to incorrect unwinding of miRNA precursors, thereby reducing the expression of mature miRNAs. MiR-466d-3p degradation induced by NS3, promotes IL-1β expression and JEV replication. Arginine molecules of NS3 are the main miRNA-binding sites. NS3 proteins of ZIKV, WNV, DENV1, and DENV2 can also degrade miRNAs.	[343]
NOD1	HCV	NS5B	NS5B increases expression of RIG-I, MDA5, TLR3, MAPK/ERK, IL-8, and TNF-α, which are accompanied by a dramatic increase in IFN-β expression in HepaRG cells. NOD1 expression is activated by NS5B, most likely through sensing dsRNA synthesized by NS5B from cellular templates. NOD1 activation is RIG-I and MAVS-independent.	[349]

## 7. Conclusions and Future Directions

An analysis of the published literature demonstrated that *Flaviviridae* nonstructural proteins are actively involved in multiple signaling cascades, including those triggering inflammation. These molecular processes often correlate with systemic inflammatory processes that drive pathogenesis of flaviviral infections. A vascular leak in endothelial cells, characteristic of the severe dengue form, is hypothesized to be triggered by DENV NS1, and by TLR4-dependent, and independent mechanisms [136,139]. Neuroinflammation caused by neurotropic flaviviruses may be significantly associated with effects of nonstructural proteins on brain cells, i.e., JEV NS3 has been demonstrated to repress inhibitory miRNA in the mouse brain, thus releasing the suppression of IL-1β secretion and promoting neuroinflammation in the brain [343]. The expression of HCV polymerase NS5B in mouse liver contributes to liver inflammation by the significant induction of IL-6, which may contribute to chronic liver inflammation and its progress to hepatocellular carcinoma in patients with chronic HCV [230].

In spite of the large number of investigations concerning *Flaviviridae* interactions with the innate immune system, there are still many blind spots in this field. One of the major hurdles in the studies of viruses and signaling pathways is the use of appropriate cell lines. Ideally, a cell line used for the study should be relevant in terms of viral replication sites, and at the same time should have unmodified functional signaling and metabolic pathways. For most *Flaviviridae*, dendritic cells appear to be a common initial target of replication [339,352,353,354,355,356]. For neurotropic viruses, such as JEV, TBEV, WNV, and ZIKV, as well as for non-neurotropic DENV, the replication in the cells of CNS (and also neuroprogenitor cells as for ZIKV) is detected [357,358,359,360,361]. ZIKV may also replicate in placenta and in the cells of the reproductive tract [362]. HCV and YFV replicate at high levels in liver cells [363,364]. Only a few studies have been performed in the cell lines that resemble the site of viral replication in vivo, because of the unavailability of cell lines, impossibility to maintain viral replication in vitro, technical challenges, etc. Hence, the results gained in the suboptimal cell lines should be translated into in vivo conceptions with much caution.

The data of the studies are often controversial, i.e., the same nonstructural protein may have opposite effects demonstrated by different labs. This phenomenon is often attributed to the differences in study models utilizing different cell lines, different ways of viral and cellular proteins expression, and different viral strains, characterized by divergent cytokine expression profiles. In addition, commercial bacterial-derived viral proteins should not be used for such studies as they may activate signaling cascades in an unspecific manner and often do not fold properly. Of note, only a few studies consider the cooperative effect of the nonstructural proteins on the components of innate immunity [266], but it would be especially interesting to study the collaborative efforts between multiple nonstructural proteins, given that they interact with each other during the viral life cycle, and these complexes may have undetermined functions. In addition, inflammasomes activation by *Flaviviridae* and their crosstalk with IFN pathways require more extensive studies, as inflammasomes may play both negative and positive roles for the pathogenesis of different flaviviral infections.

More detailed and precise data will help to settle all these arguments and to provide the progress in antiviral therapy development. Nowadays, the only licensed direct-acting antivirals (DAA) aimed at *Flaviviridae* are against HCV. They are presented by the combination of NS5A, NS5B, and NS3/4A inhibitors and are quite effective in providing a treatment response above 90% [365]. Experimental DAA against other members of *Flaviviridae* are mainly aimed at the inhibition of viral enzymes, protease, and polymerase [366]. These drugs could not only suppress viral replication but also attenuate the negative effects of viral enzymes on the host’s innate immunity cascades. Many compounds could target flavivirus enzymes, but it is difficult to transfer them into flavivirus-specific therapeutics [366]. One of the reasons for such a problem in the case of flaviviral protease is that its active site is shallow and highly charged [367]. In addition, the shared problem of most of the DAA is the emergence of drug resistant viruses, which was already noticed for HCV variants detected in HCV chronic patients [368]. In this case, the knowledge of how flaviviral proteins interfere with cellular functions has an exceptional significance. For example, the TRAF6 inhibitor peptide disrupted the complex of TRAF6, a E3 ubiquitin ligase essential for NF-κB and IFN responses, with Langat virus NS3 and could directly inhibit NS3 function and viral replication, thus helping to overcome obstacles associated with the development of DAA against NS3 [264]. DENV NS1 TLR4-dependent activity, which is hypothesized to induce endothelial vascular leak during severe dengue, could be inhibited by the use of TLR4 blocking antibodies and antagonists, which was demonstrated in vivo in a mouse model [369]. Furthermore, a number of potential treatments known to block TLR4 activation have already been trialed in humans for the treatment of other diseases [369], and thus, could be translated for the treatment of *Flaviviridae*. One more strategy to prevent DENV NS1-dependent vascular leak is the inhibition of molecules involved in glycocalyx disruption, such as sialidases, cathepsin L, and heparanase [139]. Currently licensed sialidase inhibitors for the treatment of influenza [370] and heparinase inhibitors for cancer therapy [371,372] may be used. Thus, to successfully translate more antivirals already tested in humans, for the treatment of flaviviral infections, we need to learn more about the molecular mechanisms of flaviviral proteins’ contribution to disease pathogenesis.

In view of global warming, growing freedom of movement all over the world, and the SARS-CoV-2 pandemic, which brings long-term negative effects on human health, in particular on the immune, nervous, and cardiovascular systems, *Flaviviridae* may pose a larger threat than before, and cause significant outbreaks or even a pandemic. With this scenario, the continuation of flavivirus studies, the development of antivirals, and their translation into clinical practice are extremely essential.

## Data Availability

Not applicable.

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
