# Peer review of "Flaviviridae Nonstructural Proteins: The Role in Molecular Mechanisms of Triggering Inflammation"

_viruses, 2022, doi:10.3390/v14081808_

Round 1

Reviewer 1 Report

This review is extensive and well written. While not required, some sort of graphical abstract or cartoon(s) would improve the central theme of the authors. A list of minor comments is below:

·         Line 59-60: YFV is not considered neurotropic.

·         Line 90: cycle should be changed to lifecycle

·         Line 89-96: Define viroporin activity of NS2A and NS2B

·         Line 143: word choice “for today” is unclear, maybe change to currently

·         Line 143-144: While some TLRS are conserved from humans to mice, many differ in structure or function. Please clarify this point a bit.

·         Line 229-230: “Grab the scientists attention” should be reworded.

·         Line 230 and on: Describe what species of NS1 activates immune cells. Describe the difference between intracellular and extracellular NS1.

·         Line 263: There are far more papers showing a protective role of anti-NS1 antibodies than papers demonstrating auto-reactive antibodies. Further, the role of these autoreactive antibodies in contributing to severe dengue in vivo is unclear. Please make it more apparent in the discussion and cite more papers for the protective nature of these antibodies in vivo. There are many papers showing their protective nature that are omitted in this review.

·         Line 306: “taken all around” should be changed to “taken together”

·         Table 1: TLR4 antibodies haven’t been shown to protect mice from vascular leak in vivo in citation 131. The role of TLR4 in vascular leak in vivo is highly controversial.

·         Table 1: Citation 135 doesn’t demonstrate that NS1 activates endocan expression in patients. Correlation not causation, please clarify this point.

·         For most sections it would be helpful to incorporate which cell type was used for a study and comment on the physiological relevance of such cell types. For DENV for example, it is controversial whether endothelial cells are infected in humans so the relevance of DENV infection models in endothelial cells are unclear.

·         Line 476: Should “provide” be “promote”?

·         Can the authors describe briefly how/why viruses may want to impact NFKB signaling? Proinflammatory signaling or cell proliferation state?

·         A few instances where viruses are not properly abbreviated.

·         Line spanning 563-564 needs to be re-written.

·         Line 642: “are accumulated” should be changed to “have accumulated”

Line 682 requires a reference for the WNV claim. 

Author Response

(x) Moderate English changes required

The manuscript text was edited, some sentences were rephrased and spelling mistakes etc. were corrected.

This review is extensive and well written. While not required, some sort of graphical abstract or cartoon(s) would improve the central theme of the authors.

We thank the reviewer for the positive evaluation of our manuscript and for the valuable remarks.

A list of minor comments is below:

  • Line 59-60: YFV is not considered neurotropic.

We thank the reviewer for this important remark and corrected the text of the review accordingly. Indeed, YFV is attributed to non-neurotropic virus causing visceral disease, as the main source of its replication are non-neural cells, i.e. liver cells. However, we would like to point the fact that there is evidence for YFV and, in rare cases, for its vaccine strain YFV-17D, that they may enter the CNS and are capable of causing encephalitis.

  • Line 90: cycle should be changed to lifecycle

Here and elsewhere the change was made.

  • Line 89-96: Define viroporin activity of NS2A and NS2B

This information has been added into the text.

  • Line 143: word choice “for today” is unclear, maybe change to currently

The “for today” has been changed.

  • Line 143-144: While some TLRS are conserved from humans to mice, many differ in structure or function. Please clarify this point a bit.

The necessary information has been added to the section describing TLRs.

  • Line 229-230: “Grab the scientists attention” should be reworded.

The phrase has been reformulated.

  • Line 230 and on: Describe what species of NS1 activates immune cells. Describe the difference between intracellular and extracellular NS1.

We would like to thank the reviewer for this remark, as a little introduction to flaviviral NS1 species will improve the understanding of the TLR-NS1 interactions described in the paper. The necessary information has been added to the paper.  

Line 263: There are far more papers showing a protective role of anti-NS1 antibodies than papers demonstrating auto-reactive antibodies. Further, the role of these autoreactive antibodies in contributing to severe dengue in vivo is unclear. Please make it more apparent in the discussion and cite more papers for the protective nature of these antibodies in vivo. There are many papers showing their protective nature that are omitted in this review.

We thank the reviewer for this important remark, and added more links on the protective role of anti-NS1 antibodies and changed the text appropriately.

  • Line 306: “taken all around” should be changed to “taken together”

The recommended change has been made.

  • Table 1: TLR4 antibodies haven’t been shown to protect mice from vascular leak in vivo in citation 131. The role of TLR4 in vascular leak in vivo is highly controversial.

The corresponding correction has been introduced into the text of the table.

  • Table 1: Citation 135 doesn’t demonstrate that NS1 activates endocan expression in patients. Correlation not causation, please clarify this point.

We thank the reviewer for this important remark and clarified this point in the text and in the table.

  • For most sections it would be helpful to incorporate which cell type was used for a study and comment on the physiological relevance of such cell types. For DENV for example, it is controversial whether endothelial cells are infected in humans so the relevance of DENV infection models in endothelial cells are unclear.

The missing information on the cell types was added to the tables. The information on the specificity of flaviviral replication to the cell types and discussion of cell lines relevance was also added in the “Conclusions” section.

  • Line 476: Should “provide” be “promote”?

The word has been changed as suggested.

  • Can the authors describe briefly how/why viruses may want to impact NFKB signaling? Proinflammatory signaling or cell proliferation state?

The information has been added in the text in the Section describing NF-kB signaling interference.

  • A few instances where viruses are not properly abbreviated.

The found wrong abbreviations were corrected.

  • Line spanning 563-564 needs to be re-written.

It has been re-written.

  • Line 642: “are accumulated” should be changed to “have accumulated”

The correction was made.

 Line 682 requires a reference for the WNV claim. 

The reference was added.

Reviewer 2 Report

In this manuscript, the authors review how Flaviviridae nonstructural proteins are able to trigger inflammation, going into details on the molecular mechanisms involved. This review is well-written and timely in its relevance.

Author Response

(x) English language and style are fine/minor spell check required

The manuscript text was edited, some sentences were rephrased and spelling mistakes etc. were corrected.

In this manuscript, the authors review how Flaviviridae nonstructural proteins are able to trigger inflammation, going into details on the molecular mechanisms involved. This review is well-written and timely in its relevance.

We thank the reviewer for the positive evaluation of our manuscript.